# Comparing Aluminum Concentrations in Adult and Pediatric Parenteral Nutrition Solutions: Multichamber-Bag versus Compounded Parenteral Nutrition

**DOI:** 10.3390/nu16071024

**Published:** 2024-04-01

**Authors:** David Berlana, Juan López-Hellín, Alba Pau-Parra, Roser Ferrer-Costa

**Affiliations:** 1Pharmacy Department, Vall d’Hebron Barcelona Campus Hospital, 08035 Barcelona, Spain; 2Pharmacology, Toxicology and Therapeutic Chemistry Department, Faculty of Pharmacy and Food Sciences, University of Barcelona, 08028 Barcelona, Spain; 3Biochemistry Department, Vall d’Hebron Barcelona Campus Hospital, 08035 Barcelona, Spain; 4Biochemical Chemistry, Drug Delivery & Therapy (BC-DDT) Research Group, Vall d’Hebron Institut de Recerca (VHIR), 08035 Barcelona, Spain

**Keywords:** parenteral nutrition, aluminum, multichamber-bag, compounding, safety, toxicity

## Abstract

Aluminum contamination in parenteral nutrition (PN) solutions can lead to neurotoxicity, reduced bone mass, and liver toxicity, especially in pediatric patients. Ingredients commonly used in PN compounding, such as vitamins, trace elements, calcium, and phosphate salts, contain significant amounts of aluminum. This study aimed to compare aluminum concentrations in multichamber-bag (MCB) and compounded PN for adults and pediatrics. A prospective study assessed aluminum concentrations in various types of MCB and compared them with compounded PN formulations with similar compositions. The types of MCB included Lipoflex^®^ (without electrolytes), Omegaflex^®^, Finomel^®^, Smofkabiven^®^ (with and without electrolytes), Olimel^®^, Clinimix^®^, and Numeta^®^. Overall, 80 aluminum determinations were included: 36 for MCBs and 44 for compounded PN. MCBs showed significantly lower aluminum concentrations than compounded PN: 11.37 (SD 6.16) vs. 21.45 (8.08) µg/L, respectively. Similar results were observed for adult (n = 40) and pediatric (n = 40) PN formulations (12.97 (7.74) vs. 20.78 (10.28) µg/L, and 9.38 (2.23) vs. 22.01 (5.82) µg/L, respectively). Significant differences were also found between MCBs depending on the manufacturing company. These findings suggest that MCBs PN offer a safer option for reducing aluminum contamination in PN. Harmonizing regulations concerning aluminum concentrations in PN solutions could help mitigate differences between PN formulations.

## 1. Introduction

Aluminum contamination in parenteral nutrition (PN) solutions has been documented for decades [1,2]. This can result in elevated blood aluminum, neurotoxicity, reduced bone mass, and liver toxicity [1,3,4]. Efforts to minimize the aluminum content in PN, especially in the United States, have led to the establishment of specific limits (toxic limit, upper safe limit, and unsafe limit) for aluminum intake in patients on long-term and pediatric PN [5]. The recommended limit of 25 µg/L has been widely adopted in an attempt to reduce aluminum toxicity, especially in certain patient populations. However, challenges persist, given that additives commonly used in PN compounding, such as vitamins, trace elements, calcium, and phosphate salts, all contain significant amounts of aluminum [2,6,7].

The degree of aluminum contamination in parenteral additives can be influenced by several factors, such as the source of raw materials and container design. Glass containers, in particular, have more aluminum contamination than plastic ones [1,6]. Notably, data related to calcium solutions used for PN compounding suggest that glass-packaged calcium gluconate should be avoided, and plastic-packaged calcium gluconate should be used instead, as higher aluminum contamination has been found in a glass containers of calcium compared to plastic ones [6,7].

Therefore, patients receiving plastic-packaged multichamber-bag (MCB) PN could be expected to receive lower amounts of aluminum compared to those receiving compounded PN. In this context, our recent study on aluminum concentrations in hospitalized adult patients revealed higher concentrations in patients receiving prepared PN compared to those receiving MCB PN [8]. In light of these findings, the main objective of this study was to compare aluminum concentrations in different types of PN, including MCB and compounded PN. Specifically, we quantified aluminum concentrations in commercially available MCB for adult and pediatric patients, comparing them to the aluminum concentrations in compounded PN with a similar composition. A secondary aim was to assess the aluminum content variation between different MCB formulations and compounded PN formulations to identify specific ingredients contributing to increased aluminum content in PN.

## 2. Materials and Methods

We carried out a prospective study to evaluate aluminum concentrations across various types of MCB and compared them against compounded PN with similar compositions. We also examined the variations in aluminum concentrations among different MCB as well as between compounded PN based on the source of amino acids or electrolytes.

The selected MCBs included those currently in use for adults in our hospital and those approved for pediatrics [9]. Testing involved two different batches for each MCB type, without introducing any additives. Batches were collected based on availability throughout the study period. As currently available MCBs have no vitamins or trace elements, they were not included in compounded PN either. After breaking the seals to mix the ingredients of the MCB, PN bags were refrigerated for 24 h. Since pediatric MCB is also designed to be used without mixing the lipid chamber, pediatric MCBs free of lipid were also tested. Therefore, only the glucose and amino acids chamber for each pediatric MCB were mixed, and samples were extracted identically.

In order to determine differences in aluminum content between compounded PN, we chose formulations that varied in the sources of amino acids and electrolytes. For adult formulations, one prepared PN had Aminoplasmal PO^®^ 500 mL (B Braun, Melsungen, Germany), and the other included Vamin^®^ 18 500 mL (Fresenius Kabi, Bad Homburg, Germany). The pediatric formulations involved at least one compounded PN prepared with Primene^®^ 10% 250 mL (Baxter, Deerfield, IL, USA), as an amino acid source, while the other formulation for prepared PN had Aminoven Infant^®^ 10% 250 mL (Fresenius Kabi, Bad Homburg, Germany). Similar to MCB, the PN solutions used for compounding included macronutrients and electrolytes commonly used for adult and pediatric PN in our hospital and commercially available during the study period. Regarding electrolytes, there were two options for the formulations: they could be prepared with only commercially available electrolytes, or with a combination of bulk and commercially available electrolytes. The composition of each MCB and prepared PN is listed in Table 1 and Table 2, for adults and pediatric patients, respectively.

Ingredients employed for the compounded PN included:-Amino acids:
-For adults: Aminoplasmal PO^®^ 12.5% 500 mL (BBraun, Melsungen, Germany) or Vamin^®^18 (Fresenius Kabi, Bad Homburg, Germany), both in a glass container.-For pediatric patients: Primene^®^ 10% 250 mL (Baxter, Deerfield, IL, USA) or Aminoven Infant^®^ 10% 250 mL (Fresenius Kabi, Bad Homburg, Germany), both in a glass container.
-Glucose: Glucose 70% 500 mL (Baxter, Deerfield, IL, USA) in a plastic container.-Lipids: Smoflipid^®^ 20% 500 mL (Fresenius Kabi, Bad Homburg, Germany) in a glass container.-Electrolytes:
-Potassium: ClK 2 M 50 mL (bulk product, Farmacia Carreras, Barcelona, Spain) in a glass container or ClK 2 M 10 mL (BBraun, Melsungen, Germany) in a plastic container.-Sodium: NaCl 20% 10 mL (BBraun, Melsungen, Germany) in a plastic container.-Magnesium: Magnesium sulfate 50 mL (bulk product, Farmacia Carreras, Barcelona, Spain) or Magnesium sulfate 10 mL (BBraun, Melsungen, Germany; commercial product), both in a glass container (no plastic container available).-Calcium: calcium gluconate 50 mL (bulk product, Farmacia Carreras, Barcelona, Spain) in a glass container, or calcium gluconate Suplecal^®^ 10 mL (BBraun, Melsungen, Germany) in a plastic container.-Phosphate: Glycerophosphate 10 mL (Fresenius Kabi, Bad Homburg, Germany) in a plastic container.
-Water for injection: Water for injection 500 mL (Fresenius-Kabi, Bad Homburg, Germany) in a glass container.

The compounded PN solutions were also stored in a refrigerator for 24 h after being prepared. The 24 h storage period was chosen to align with the regular duration of PN administration. Additionally, it is common practice for hospital pharmacies to compound both compounded PN and MCB in advance and refrigerate the PN bags, especially over the weekend, to ensure they are readily available for administration. For aluminum analysis, 1 mL samples were uniformly extracted via the bag injection ports using a syringe (BD Emerald^®^, [Becton Dickinson, Fraga, Spain]) with a needle (BD^®^ Blunt fill, [Becton Dickinson, Fraga, Spain]) after this 24 h period for both types of PN (MCB and compounded PN). The compounded PN solutions were prepared using an automated compounder (ExactaMix^®^ 2400, Deerfield, IL, USA). Therefore, the company providing the compounder supplied consumables used for PN preparation, including valve sets, filling tubes, and bags. All aluminum tests for adult formulations were conducted on the same day in 2022 (22 June), whereas tests for pediatric formulations were carried out on a different day in 2022 (21 September), according to the same circumstances and methods.

For aluminum determination, samples were diluted 1:40 in 1% nitric acid, and aluminum was measured using inductively coupled plasma–mass spectrometry (Nexion 350X instrument; PerkinElmer, Waltham, MA, USA) with scandium as an internal standard. The method was validated according to the European Medicines Agency (EMA) guidelines. Aluminum control material was obtained from ClinCheck^®^ Recipe (Chemicals & Instruments GmbH, Munich, Germany) in line with the German Medical Association on Quality Assurance. The accuracy and precision of the method were verified using the following certified reference material for calibration: Multi-element Calibration Standard 3 PerkinElmer^®^ (certified by EURACHEM ISO 17025).

Descriptive statistics [mean (standard deviation—SD)] were carried out for aluminum test results. Differences in aluminum content between MCB formulations and compounded PN solutions were analyzed using Student’s *t*-test or the corresponding nonparametric test (Mann–Whitney U test). Differences in aluminum assay between the different types of MCBs and between the different formulations of compounded PN solutions were analyzed using one-way ANOVA, the post hoc multiple-comparison Bonferroni test, and the nonparametric Kruskal–Wallis test. All statistical analyses were performed with Stata software, release 16.0 (Stata Corporation, Lakeway, TX, USA); and statistical significance was set at *p* < 0.05.

## 3. Results

Overall, 80 aluminum determinations were included in the study: 36 for MCBs and 44 for compounded PN. Aluminum concentrations from MCBs were significantly lower than those from compounded PN: 11.37 (standard deviation [SD] 6.16) vs. 21.45 (8.08) µg/L, respectively (Table 3). Similar values of aluminum concentration were found in MCB and compounded PN in adult formulations without electrolytes (10.49 [9.60] and 7.64 [3.89] µg/L, respectively, Table 3). Differences in aluminum content in MCB formulations and compounded PN classified into different groups are listed in Figure 1.

In general, at least two different batches for each MCB were assessed for aluminum concentrations (four different batches for Numeta^®^ G16 (Baxter, Deerfield, IL, USA) and three for all adult MCB with lipid and electrolytes), except for Clinimix^®^ with electrolytes (Clinimix^®^ N12GE and Clinimix^®^ N6GE—Baxter, Deerfield, IL, USA) with only one batch for each formulation due to the lack of availability. Regarding the type of MCB, 2-in-1 formulations (glucose + amino acids) showed lower content of aluminum (3.20 [1.20] and 1.95 [0.20] µg/L for Clinimix^®^ with and without electrolytes, respectively, Figure 1). However, statistical significance was not achieved, likely due to the limited number of Clinimix^®^ samples. Despite the fact that MCB Lipoflex Lipid Special^®^ (BBraun, Melsungen, Germany) has no electrolytes, it showed a higher content of aluminum compared to the other MCBs (21.20 [8.63] µg/L; Figure 1). Kruskal–Wallis tests showed significant differences between Omegaflex^®^—BBraun, Melsungen, Germany- (23.47 [2.99] µg/L) and the following MCB: Finomel^®^ -Baxter, Deerfield, IL, USA- (9.69 [1.72] µg/L), Smofkabiven^®^ -Fresenius Kavi, Bad Homburg, Germany- (1.82 [1.00] µg/L), Olimel^®^ -Baxter, Deerfield, IL, USA- (17.36 [1.72] µg/L), Numeta^®^ (9.38 [2.23] µg/L) and the mean of the rest of MCB with lipid and electrolytes for adults (17.88 [4.96] µg/L). Similarly, the lower aluminum content in MCB was found in Finomel^®^, showing a significant difference compared to Omegaflex^®^, Smofkabiven^®^, Olimel^®^, and the mean of the rest of MCBs with lipid and electrolytes for adults (Figure 1). Regarding compounded PN, the formulation corresponding to Omegaflex^®^ showed the higher aluminum concentration (Figure 1).

With regard to differences within compounded PN, differences have been shown depending on the source of amino acids used in both adult and pediatric formulations (Table 4. Notably, formulations prepared with Aminoplasmal PO^®^ 12.5% for adults and Aminovent Infant^®^ showed higher aluminum concentrations (Table 4). Identically, formulations with only commercial electrolytes showed lower content of aluminum compared to formulations containing bulk electrolytes: 20.90 (4.90) vs. 24.89 (6.36) µg/L, respectively (Table 4).

## 4. Discussion

Our results revealed that MCBs contain lower aluminum concentrations across various categories compared to compounded PN. Both adult and pediatric MCBs consistently demonstrate reduced content compared to their equivalent compounded PN. However, significant variations were observed based on the manufacturing company in MCB, as well as in the specific components used in compounded PN.

In general, all MCBs showed lower aluminum concentrations than compounded PN, except for electrolyte-free MCBs with lipids. These findings align with our prior results showing lower aluminum blood concentrations in adult patients receiving only MCB compared to those also receiving compounded PN [8]. Additionally, our results confirm the role of electrolytes in aluminum load, as MCBs without electrolytes showed lower concentrations than those with electrolytes [1,10]. As expected, compounded PN without electrolytes also showed lower concentrations of aluminum than those with electrolytes. Notably, all MCB formulations contained calcium in the form of calcium chloride, whereas compounded PN solutions were prepared with calcium gluconate (this product is available commercially or in bulk packaging). This fact might contribute to the lower aluminum concentration in MCB, since calcium gluconate has been associated with higher aluminum content than the equivalent form of calcium chloride [6,7,11,12].

Our findings identified differences between MCBs (Figure 1), suggesting that, in addition to electrolytes, macronutrients might also play an important role in aluminum content within PN solutions. When comparing MCB formulations for adults containing lipid and electrolytes, we observed that those with lower concentrations of macronutrients (mainly amino acids) and electrolytes, such as Finomel^®^ and Smofkabiven^®^, exhibited lower aluminum concentrations (9.69 [1.15] and 12.82 [1.00] µg/L, for Finomel^®^ and Smofkabiven^®^, respectively). In contrast, MCB formulations with higher concentrations of macronutrients and electrolytes, like Olimel N9E^®^ and Omegaflex especial^®^, showed higher aluminum concentrations (17.36 [1.72] and 23.47 [2.99] µg/L, for Olimel^®^ and Omegaflex^®^, respectively). The load of aluminum might also be due to processing methods in different companies, since the higher aluminum concentration was detected in two MCBs from the same company. These findings are in agreement with a recent work showing Omegaflex^®^ with and without electrolytes had higher aluminum concentrations [13]. Differences between MCB may also be explained by the source of lipids, especially in MCB without electrolytes, since the lipid content in all-in-one PN has shown wide variation in its aluminum concentration [13,14,15]. However, the study of Ozturk et al. did not identify the type of lipid or the type of all-in-one PN tested.

As mentioned above, differences were also found between MCBs depending on the manufacturing company, suggesting that raw materials and processing factors might play a pivotal role in aluminum load within PN solutions. This point is in line with prior studies showing wide differences in aluminum concentrations between different products [14,16]. Moreover, it was also supported by the results obtained for compounded PN based on the source of amino acids, both in pediatric and adult patients. These results showed higher concentrations of aluminum in adult formulations with Aminoplasmal PO^®^ 12.5% compared to their equivalents prepared with Vamin^®^18. Similarly, compounded pediatric PN with Primene^®^ achieved lower aluminum content than pediatric PN prepared with Aminoven Infant^®^ (Table 4). These findings align with prior studies, suggesting that, after micronutrients (electrolytes, vitamins and trace elements), amino acids might be the main source of aluminum content in PN solutions [17,18,19].

While none of the MCBs exceeded the widely accepted aluminum limit of 25 µg/L, it is noteworthy that Lipoflex^®^ Lipid Special without electrolytes and Omegaflex^®^ approached this threshold. This observation is significant, since these MCBs achieved these values without the addition of vitamins or trace elements, which are known to contribute to aluminum load [1,7,14]. Therefore, next steps to identify the origin of the aluminum contamination should be taken to implement actions to reduce aluminum content in these MCB and PN solutions in general.

Pediatric formulations also exhibited significantly lower aluminum concentrations in MCB compared to compounded PN. Pediatric compounded PN achieved values close to the accepted limit of 25 µg/L, particularly without the addition of vitamins or trace elements [5]. Consequently, higher amounts of aluminum and overcoming the limit of 25 µg/L would be expected after the addition of vitamins and trace elements. This finding is particularly noteworthy, considering that pediatric patients are more susceptible to aluminum toxicity [1,20,21].

The contribution of electrolytes when compounding PN has been demonstrated in several works [1,10,11,17]. Our findings align with this point, as formulations with electrolytes in compounded PN showed significantly lower aluminum concentrations than equivalent formulations with electrolytes (Table 4: p.eg. 22.11 (2.29) vs. 7.73 (1.52) µg/L for formulations of Smofkabiven^®^ with and without electrolytes, respectively). Compounded PN formulations without bulk-packaged electrolytes demonstrate lower aluminum concentrations, reinforcing the importance of container material [6,7].

This study emphasizes the need for clinicians to consider not only electrolytes but also macronutrients and packaging choices to mitigate aluminum exposure in PN solutions. Therefore, manufacturing companies should explore strategies to minimize aluminum contamination within PN solutions, as their own MCBs showed lower aluminum content than equivalent compounded PN formulations. These insights hold clinical relevance for both adult and pediatric patients, guiding the selection of PN solutions to minimize the risk of aluminum toxicity. Consequently, products with the lowest concentrations should be preferred for use. The variability in aluminum concentrations found in this study, as well as in prior works, stresses the importance of defining quality standards for PN solutions. Our findings showed a high SD (Table 3) due to the variability in aluminum concentration among the different MCBs and PN solutions used. However, the SD was lower when MCBs or compounded PN with similar compositions were grouped together (Figure 1 and Table 4). Considering that no physiological role is expected for aluminum, we might consider low aluminum infusions as safe, such as suggested by the ASPEN (up to 2 µg/kg body weight/day) [22]. However, a lower limit would be preferred, especially in pediatrics and long-term PN, since aluminum toxicity has been described in pediatric patients despite regulations [20,21]. Therefore, limits for aluminum in PN solutions should consider the total amount of aluminum received by the patient, and not only indicate the maximum amount of aluminum per liter, as patients may receive several PN solutions.

To our knowledge, this is the first study comparing concentrations of aluminum in MCBs to their equivalent compounded PN formulations in both pediatrics and adults. However, this study has some limitations that might restrict the interpretations of our results. The first limitation is the limited type of MCB tested, as aluminum determination was only performed in adult MCB formulations included in our hospital protocol. Furthermore, only 3 types of MCB formulation without electrolytes were included, limiting the assessment of the role of electrolytes and type of electrolytes included within MCB in aluminum contamination. The second limitation is related to the low number of tests performed and the lower number of batches per type of MCB tested. Despite this fact, the low value of the standard deviation shown in the results of the MCBs might reflect the low variability between batches of the same MCB formulation. In addition, our findings are in agreement with recent results assessing the aluminum concentration in various MCB [13]. Besides, all commercially available MCBs for pediatric patients were tested, including at least two batches for each pediatric formulation. The third limitation is related to the products selected to elaborate the compounded PN. Although it is known that glass containers are linked to aluminum contamination, the study was performed by using Smoflipid^®^ in a glass container for all compounded PN with lipids. This fact might limit the validity of the comparation with compounded PN. Nevertheless, a low difference has been found between types of lipids [14,17]. Another limitation could be the method used to assess aluminum in PN solutions. Currently, there is no standard method for quantifying aluminum in PN emulsions. However, the inductively coupled plasma–mass spectrometry method has already been used in prior studies in PN emulsions [13,23].

## 5. Conclusions

In conclusion, our findings support the preference for MCB products, especially in home PN settings and pediatrics, due to consistently lower aluminum concentrations. Future research should focus on identifying the sources of aluminum contamination to implement effective mitigation strategies. Additionally, harmonizing regulations concerning aluminum concentrations in PN solutions could help mitigate differences between PN formulations and PN products from different companies.

## Figures and Tables

**Figure 1 nutrients-16-01024-f001:**
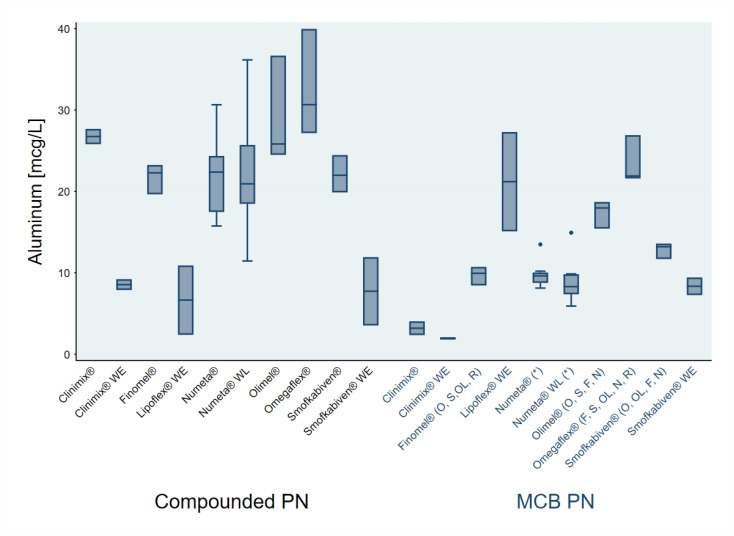
Aluminum content in MCB formulations and the corresponding compounded PN. MCB: multichamber-bag. PN: parenteral nutrition. F: significant difference compared to Finomel^®^; N: significant difference compared to Numeta^®^; O: significant difference compared to Omegaflex^®^; OL: significant difference compared to Olimel^®^; R: significant difference compared to the rest of MCB with lipid and electrolytes; S: significant difference compared to Smofkabiven^®^ with electrolytes. * Significant difference compared to MCB aluminum content (*p* < 0.05).

**Table 1 nutrients-16-01024-t001:** Multichamber-bag formulations for adults and compounded PN composition.

MCB Formulation (Content per 1 L)	MCB	Compounded PN (Different Amino Acid Source—at Least 1 Sample from Each Formulation)
Amino acids 56.0 g: Carbohydrate 144.0 g; Lipid 40 g; No electrolytes	Nutriflex lipid special^®^ without electrolytes	Aminoplasmal PO^®^ 12.5% Vamin^®^18
Amino acids 56.0 g; Carbohydrate 144.0 g; Lipid 40 g Sodium 40.0 mmol; Potassium 28.0 mmol Magnesium 3.2 mmol; Calcium 3.2 mmol Phosphate 12.0 mmol	Omegaflex^®^ special	Aminoplasmal PO^®^ 12.5% + bulk electrolytes Aminoplasmal PO^®^ 12.5% + commercial electrolytes Vamin^®^18 + bulk electrolytes
Amino acids 51.0 g; Carbohydrate 127.0 g Lipid 38 g; No electrolytes	Smofkabiven^®^ without electrolytes	Aminoplasmal PO^®^ 12.5% Vamin^®^18
Amino acids 51.0 g; Carbohydrate 127.0 g Lipid 38 g Sodium 41.0 mmol; Potassium 30.0 mmol; Magnesium 5.1 mmol; Calcium 2.5 mmol; Phosphate 13.0 mmol	Smofkabiven^®^	Aminoplasmal PO^®^ 12.5% + bulk electrolytes Aminoplasmal PO^®^ 12.5%+ commercial electrolytes Vamin^®^18 + bulk electrolytes
Amino acids 56.9 g; Carbohydrate 110.0 g Lipid 40 g Sodium 35.0 mmol; Potassium 30.0 mmol; Magnesium 4.0 mmol; Calcium 3.5 mmol; Phosphate 15.0 mmol	Olimel^®^N9E	Aminoplasmal PO^®^ 12.5% + bulk electrolytes Aminoplasmal PO^®^ 12.5% + commercial electrolytes Vamin^®^18 + bulk electrolytes
Amino acids 50.6 g; Carbohydrate 126.9 g Lipid 40.1 g Sodium 40.6 mmol; Potassium 30.5 mmol; Magnesium 5.1 mmol; Calcium 2.6 mmol: Phosphate 12.7 mmol	Finomel^®^	Aminoplasmal PO^®^ 12.5% + bulk electrolytes Aminoplasmal PO^®^ 12.5% + commercial electrolytes Vamin^®^18 + bulk electrolytes
Amino acids 35.0 g; Carbohydrate 100.0 g No electrolytes	Clinimix^®^ N12G20	Aminoplasmal PO^®^ 12.5% Vamin^®^18
Amino acids 35.0 g; Carbohydrate 100.0 g Sodium 35.0 mmol; Potassium 30.0 mmol Magnesium 2.5 mmol; Calcium 2.3 mmol; Phosphate 15.0 mmol	Clinimix^®^ N12G20E	Aminoplasmal PO^®^ 12.5% + bulk electrolytes Vamin^®^18 + bulk electrolytes
Amino acids 28.0 g; Carbohydrate 75.0 g Sodium 35.0 mmol; Potassium 30.0 mmol; Magnesium 2.5 mmol; Calcium 2.3 mmol Phosphate 15.0 mmol	Clinimix^®^ N9G15E	Aminoplasmal PO^®^ 12.5% + bulk electrolytes Vamin^®^18 + bulk electrolytes

**Table 2 nutrients-16-01024-t002:** Multichamber-bag formulations for pediatrics and compounded PN composition.

MCB Formulation (Content per 1 L)	MCB	Compounded PN (Different Amino Acid Source—at Least 1 Sample from Each Formulation)
Amino acids 39.0 g; Carbohydrate 167.0 g Sodium 27.0 mmol; Potassium 26.0 mmol; Magnesium 2.0 mmol; Calcium 16.0 mmol Phosphate 13.0 mmol	Numeta^®^ G13 (without lipid)	Primene^®^ + bulk electrolytes Aminoven Infant^®^ + bulk electrolytes Primene^®^ + commercial electrolytes
Amino acids 31.0 g; Carbohydrate 133.0 g Lipid 25 g Sodium 22.0 mmol; Potassium 21.0 mmol; Magnesium 1.3 mmol; Calcium 13.0 mmol Phosphate 13.0 mmol	Numeta^®^ G13	Primene^®^ + bulk electrolytes Aminoven Infant^®^ + bulk electrolytes Primene^®^ + commercial electrolytes
Amino acids 35.0 g; Carbohydrate 206.0 g Sodium 31.0 mmol; Potassium 30.0 mmol; Magnesium 4.1 mmol; Calcium 8.2 mmol; Phosphate 8.5 mmol	Numeta^®^ G16 (without lipid)	Primene^®^ + bulk electrolytes Aminoven Infant^®^ + bulk electrolytes Aminoven Infant^®^ + commercial electrolytes
Amino acids 26.0 g; Carbohydrate 155.0 g Lipid 31.0 g Sodium 24.0 mmol; Potassium 23.0 mmol; Magnesium 3.1 mmol; Calcium 6.2 mmol Phosphate 8.7 mmol	Numeta^®^ G16	Primene^®^ + bulk electrolytes Aminoven Infant^®^ + bulk electrolytes Primene^®^ + commercial electrolytes
Amino acids 30.0 g; Carbohydrate 247.0 g Sodium 58.0 mmol; Potassium 41.0 mmol; Magnesium 3.3 mmol; Calcium 5.0 mmol; Phosphate 8.3 mmol	Numeta^®^ G19 (without lipid)	Primene^®^ + bulk electrolytes Aminoven Infant^®^ + bulk electrolytes Primene^®^ + commercial electrolytes
Amino acids 23.0 g; Carbohydrate 192.0 g Lipid 28.1 g Sodium 45.8 mmol; Potassium 32.0 mmol; Magnesium 2.6 mmol; Calcium 3.8 mmol; Phosphate 9.4 mmol	Numeta^®^ G19	Primene^®^ + bulk electrolytes Aminoven Infant^®^ + bulk electrolytes Primene^®^ + commercial electrolytes
Amino acids 39.0 g; Carbohydrate 167.0 g Sodium 27.0 mmol; Potassium 26.0 mmol Magnesium 2.0 mmol; Calcium 16.0 mmol Phosphate 13.0 mmol	Numeta^®^ G13 (without lipid)	Primene^®^ + bulk electrolytes Aminoven Infant^®^ + bulk electrolytes Primene^®^ + commercial electrolytes

**Table 3 nutrients-16-01024-t003:** Aluminum content in parenteral nutrition solutions (MCB vs. compounded).

	Multichamber-Bag	Compounded PN	*p* Value
All formulations (adult and pediatric)	11.37 (6.16) [n = 36]	21.45 (8.08) [n = 44]	<0.01
Adult formulations (all)	12.97 (7.74) [n = 20]	20.78 (10.28) [n = 20]	<0.05
With electrolytes	14.03 (6.94) [n = 14]	26.41 (5.90) [n = 14]	<0.01
Without electrolytes	10.49 (9.60) [n = 6]	7.64 (3.89) [n = 6]	ns
With lipid	15.57 (6.30) [n = 16]	21.57 (10.41) [n = 16]	<0.05
With lipid and electrolytes	15.84 (5.64) [n = 12]	26.36 (6.40) [n = 12]	<0.01
Without lipid (2-in-1 formulations)	2.58 (1.01) [n = 4]	17.54 (10.55) [n = 4]	<0.05
Pediatric formulations (all)	9.38 (2.23) [n = 16]	22.01 (5.82) [n = 24]	<0.01
With lipid	9.80 (1.64) [n = 8]	22.01 (4.99) [n = 12]	<0.01
Without lipid (2-in-1 formulations)	8.97 (2.75) [n = 8]	22.00 (6.78) [n = 12]	<0.01

Data expressed as µg/L (standard deviation) [n]. ns: not significant. MCB: multichamber-bag. PN: parenteral nutrition.

**Table 4 nutrients-16-01024-t004:** Differences in aluminum content in compounded PN depending on the source of amino acids and electrolytes.

Type of Formula	Different Source of Amino Acids	
	Aminoplasmal PO^®^ 12.5%	Vamin^®^18
Adult formulations (all)	23.27 (9.51) [n = 12]	17.05 (10.89) [n = 8]
Without electrolytes	10.67 (1.37) [n = 3]	4.60 (2.89) [n = 3]
With electrolytes	27.47 (6.67) [n = 9]	24.52 (4.17) [n = 5]
With bulk electrolytes	29.30 (8.79) [n = 5]	24.52 (4.17) [n = 5]
	Primene®	Aminoven Infant®
Pediatric formulations (all)	20.26 (5.82) [n = 12]	23.75 (5.51) [n = 12]
With bulk electrolytes	21.68 (6.53) [n = 8]	25.57 (4.61) [n = 8]
With commercial electrolytes	17.42 (2.97) [n = 4]	20.10 (5.91) [n = 4]
	Different source of electrolytes	
	Commercial electrolytes	Bulk electrolytes
All formulations	20.90 (4.90) [n = 12]	24.89 (6.36) [n = 26]
With lipid	22.73 (3.94) [n = 8]	24.91 (5.83) [n = 16]
Without lipid	17.24 (5.00) [n = 4]	24.85 (5.83) [n = 10]
Adult formulations	25.18 (1.70) [n = 4]	26.91 (6.96) [n = 10]
With lipid and Aminoplasmal PO® 12.5%	25.18 (1.70) [n = 4]	29.70 (10.10) [n = 4]
Pediatric formulations	18.77 (4.56) [n = 8]	23.62 (5.82) [n = 16]
With Primene®	17.43 (2.97) [n = 4]	21.68 (6.53) [n = 8]
With Aminoven Infant®	20.11 (5.91) [n = 4]	25.57 (4.61) [n = 8]

Data expressed as mcg/L (standard deviation) [n].

## Data Availability

The data presented in this study are available on request from the corresponding author. The data are not publicly available due to privacy reasons

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
