# Peer review of "Comparing Aluminum Concentrations in Adult and Pediatric Parenteral Nutrition Solutions: Multichamber-Bag versus Compounded Parenteral Nutrition"

_nutrients, 2024, doi:10.3390/nu16071024_

Round 1

Reviewer 1 Report

Comments and Suggestions for Authors

In this manuscript, Berlana et al. describe the measurement of aluminum concentration in adult and pediatric parenteral nutrition solutions. They compare aluminum concentration in multichamber bags from different manufacturers and compounded PN. Accumulation of aluminum concentration above safe limit leads to neurotoxicity, an effect that is particularly deleterious in children. Although of relevance, I have few point of criticisms that need to be revised.

1.         The authors should state the allowed range of Aluminum concentration in PN solution directly in the introduction and not only in the discussion.

2.         The rationale of the storage of MCB in refrigerator for 24h is unclear. MCB are prepared and immediately applied. The application time is usually between 8 and 24 hours at RT.

3.         Why do the author analyzed only glucose 70% from plastic bottles and not from glass flasks? Why did they not consider a further manufacturer? The same question regards Smoflipid and Magnesium.

4.         The SD described in the section results is high. Do the authors have an explanation for the reported data?

5.         The representation of Data shown Table 3 could be improved

6.         The discussion is very long and can be shortened.  

Comments on the Quality of English Language

Minor editing of English language required

Author Response

We thank the reviewer for your time spent reviewing our work and your valuable feedback on our manuscript. In what follows the reviewers’ comments are in black and the author’s responses are in red. A file containing the revision with tracked changes has been added to the submission. Please see the attachment. 

Reviewer 2 Report

Comments and Suggestions for Authors

The authors performed experiments on the aluminium contents of total parenteral nutrition in a wide variety of marketed formulations and components, either industrially made in multi-chamber bags or locally compounded. They address the question of the source of aluminium content and discuss the importance of European regulations.

Some minor remarks are as follow:

Lines 14f: The first sentence of the abstract starts with a quite obvious statement. However, I would have expected the authors to be more explicit on who or what experiences the adverse effects. I would encourage to mention the patient receiving parenteral nutrition not only as an implication.

Line 28 and Conclusions: There is still some points to discuss until a meaningful regulation can be set in place. What would the authors suggest to regulate, the individual Al content of the compounding components or the maximum of the final formulation? Shall there be measurements taken by quality control labs and hospital pharmacies? Does this seem feasible with instrumentation commonly present? I would also encourage to discuss the risks of maximum values of the individual compounds (overestimation of Al content in the final product and the implications for measurements in the individuals components manufacturing).

Line 34: The authors state the contamination with Al was documented for decades. Was this mostly a local observation or do they refer to literature works?

Line 46: The phrase "higher aluminium contamination" has no comparison. Probably, "relevantly elevated aluminium contamination" would transport the message better. I would encourage checking this or rephrasing.

Line 65: What about trace elements like zinc and selenium?

Table 1/2: The header of the third column end on a "]" which serves no obvious purpose. Please remove or explain.

Line 113: Were the same consumables used throughout the study? Glass or plastic containers? What kind of syringe and what kind of needle? This could have influenced the results greatly and must therefore be disclosed. Please add this to the materials.

129: Which Fisher test was used alongside the t-test? What were, explicitly enumerated, the "corresponding nonparametric test"?

Table 4: I would like to encourage the authors to consider changing table 4 into a graph (eg, a point and range chart or a dot-boxplot). The illustration might help the insterested reader to figure out the observed differences and their variations more easily.

Comments on the Quality of English Language

Line 38 and line 55: The term pediatrics seems a little wide. Does this refer to the whole field of pediatrics or pediatric PN?

Line 57: The word "variations" might be intended in singular form?

Line 114: The phrase "same day throughout 2022" is not clear. To avoid confusion, please check and state whether it was the same weekday or one particular day? Why would "throughout 2022" be used, if it had been one day?

Line 126: The header for statistical analysis is redundant since it is clear from the text and no further headers were used in the material and methods section.

Line 139: The conjunction with "However" seems to weight the statement and is unfitting for the results section.

Author Response

(The authors gave the same response as above.)

Reviewer 3 Report

Comments and Suggestions for Authors

This top article shows the amount of Al in industrially pre-mixed parenteral nutrition, allowing for a better choice when treating patients (especially children) over a longer period. The study was performed independently, despite a conflict of interest. I have some remarks:

Results: Table 3 & 4: MCB, does this include both 2-chamber and 3-chamber bags? Was there no difference between them?

Methods: For multiple comparisons, why did you use Scheffé's test instead of the standard Tukey's test?

Author Response

(The authors gave the same response as above.)
